# Vision-Based In-Flight Collision Avoidance Control Based on Background Subtraction Using Embedded System

**DOI:** 10.3390/s23146297

**Published:** 2023-07-11

**Authors:** Jeonghwan Park, Andrew Jaeyong Choi

**Affiliations:** 1ThorDrive, 165, Seonyu-ro, Yeongdeungpo-gu, Seoul 07268, Republic of Korea; jhpark@thordrive.a; 2School of Computing, Dept. of AI-SW, Gachon University, 1342 Seongnam-daero, Sujeong-gu, Seongnam 13306, Republic of Korea

**Keywords:** unmanned aerial vehicle, collision avoidance, trajectory estimation, feature-point matching, optical flow, background subtraction

## Abstract

The development of high-performance, low-cost unmanned aerial vehicles paired with rapid progress in vision-based perception systems herald a new era of autonomous flight systems with mission-ready capabilities. One of the key features of an autonomous UAV is a robust mid-air collision avoidance strategy. This paper proposes a vision-based in-flight collision avoidance system based on background subtraction using an embedded computing system for unmanned aerial vehicles (UAVs). The pipeline of proposed in-flight collision avoidance system is as follows: (i) subtract dynamic background subtraction to remove it and to detect moving objects, (ii) denoise using morphology and binarization methods, (iii) cluster the moving objects and remove noise blobs, using Euclidean clustering, (iv) distinguish independent objects and track the movement using the Kalman filter, and (v) avoid collision, using the proposed decision-making techniques. This work focuses on the design and the demonstration of a vision-based fast-moving object detection and tracking system with decision-making capabilities to perform evasive maneuvers to replace a high-vision system such as event camera. The novelty of our method lies in the motion-compensating moving object detection framework, which accomplishes the task with background subtraction via a two-dimensional transformation approximation. Clustering and tracking algorithms process detection data to track independent objects, and stereo-camera-based distance estimation is conducted to estimate the three-dimensional trajectory, which is then used during decision-making procedures. The examination of the system is conducted with a test quadrotor UAV, and appropriate algorithm parameters for various requirements are deduced.

## 1. Introduction

Lately, advancements in machine learning methodologies have enabled the development of vision-based spatial and object recognition systems, and this led to active research in the field of autonomous flight control systems, especially for unmanned aerial vehicles (UAVs). However, currently available autonomous flight control systems for UAVs focus mostly on waypoint-based global positioning systems (GPS), although the ability to navigate complex and dynamic environments is under active development [1,2]. Several drone manufacturers have integrated into their quadrotors simple autonomous flight systems that can follow subjects while avoiding static obstacles at low flight speeds, but they lack the ability to handle sudden changes in environments, and the standard function of static obstacle avoidance is still prone to failure.

The recent advances in unmanned aerial vehicles (UAVs) and computing technologies (i.e., artificial intelligence, embedded systems, soft computing, cloud and edge computing, sensor fusion, etc.) have expanded the potential and extended the capabilities of UAVs [3,4]. One of the key abilities required for autonomous flight systems is the methodology of mid-air collision avoidance [5]. Small-scale UAVs are exposed to various threat sources such as birds and other small aircrafts. Mid-air collisions almost certainly lead to vehicle damage and payload losses. An autonomous collision avoidance system capable of detecting potential hostile objects and performing decision-making avoidance maneuvers is regarded as a crucial component of an autonomous flight system [6]. 

Collision avoidance systems require a means by which to perceive potential obstacles, and this is performed by one or more types of sensors. Typical sensors employed for obstacle recognition include passive sensors such as cameras and active sensors such as RADAR, LiDAR, and SONAR. Cameras, commonly utilized passive sensors, can further be classified into monocular, stereo, and event-based types [7,8,9,10,11]. Cameras typically benefit from their small size and ease of use, but they are sensitive to lighting conditions. Lee et al. [12] employed inverse perspective mapping for object recognition, with the downside of relatively slow operational flight speeds. Haque et al. [13] implemented a fuzzy controller with stereo cameras for obstacle avoidance. Falanga et al. [14] demonstrated low-latency collision avoidance systems with bio-inspired sensors called event cameras. These sensors require no additional image processing operations, making them ideal for resource-constrained environments such as UAVs. However, event cameras are still in their active development, and their prices are still too high for the mass deployment.

Active sensors emit energy waves that reflect off object surfaces and measure distances based on the round-trip times of these waves. They are robust to lighting conditions and have a relatively broad operational range but are usually heavier and less easy to mount than passive sensors. Kwag et al. [15] utilized RADAR sensors to obtain position and velocity information about nearby objects and execute appropriate maneuvers based on this information. Moses et al. [16] developed a prototype of an X-band RADAR sensor for obstacle recognition from UAVs.

Mid-air collision avoidance systems are constrained with several requirements. Given that they operate in very short time periods, rapid cognition and response are crucial. Therefore, the computational complexity of the algorithms they use must be considered. In addition, it is desirable to utilize existing sensors such as cameras rather than using additional sensors to minimize the impact on the payload capacity.

Camera sensors benefit from their small size and low power consumption and provide an abundance of real-world information. However, this information is often highly abstracted, and additional processing is required to obtain them. One of the key objectives of vision-based perception algorithms is to require the lowest possible computational power. Furthermore, modern vision-based perception systems do not yet have robust general-inference capabilities and require further development.

There are various methodologies for moving object detection from image and video data, including statistical, image geometry-based, and machine-learning-based methods. Lv et al. [17] solved the problem of low accuracy and slow speed of drone detection by combining background difference and lightweight network SAG-YOLOv5s. The detection performance of the proposed method is 24.3 percentage higher than that of YOLOv5s, and the detection speed in 4K video reached 13.2 FPS. The proposed method achieved not only a higher detection accuracy but also a higher detection speed. However, the proposed method is only able to detect a drone under a fixed camera. Chen et al. [18] compared the motion vectors of moving and stationary objects to detect moving objects. They presented an object-level motion detection from a freely moving camera using only two consecutive video frames. However, due to the complexity of the computation, the proposed method is limited to real-time application. Kim et al. [19] clustered optical flow vectors with K-nearest neighbors clustering and distinguished moving objects based on the cluster variance. Seidaliyeva et al. [20] addressed moving object detection based on background subtraction, while the classification is performed using CNN. However, the main limitation of the proposed detector is the dependence of its performance on the presence of a moving background. 

This research aims to develop and demonstrate a moving object perception and decision-making system for reactive mid-air collision avoidance with low-cost UAV system based on background subtraction method. The concept of applying the proposed system is illustrated in Figure 1. 

The system utilizes an RGB-D camera to perform image geometry-based moving object detection. To meet the low latency requirements of collision avoidance systems, a detection algorithm with low computational complexity is devised. The novelty of the proposed system lies in the 2-D perspective transformation approximation of background motion. The corresponding points between images that are needed for approximation calculation are collected using optical flow, and to ensure that the perspective transform model best approximates the background motion, measures such as limiting optical flow estimation regions are employed. These components enable vision-based moving object detection with low computational requirements. The approximated background motion model is utilized for background subtraction to extract regions in which a moving object is present. To cope with inevitable noise originating from the approximation process and the visual data per se, various image filters and a binarization strategy are utilized. Custom-made clustering and tracking modules perceive and track individual objects for threat assessment. The distance to an object is measured by a stereo camera and then used to estimate the object’s three-dimensional trajectory relative to the ego-vehicle. This trajectory information is utilized for decision-making, triggering various commands that allow the ego-vehicle either to avoid the object or perform an emergency stop.

To test the system, a quadrotor UAV equipped with a Raspberry Pi 4 low-power computer and a low-cost Intel RealSense D435i RGB-D camera with a visual odometry camera are mounted on the platform. The proposed hardware system is shown in Figure 2. Evasive maneuvers from various conditions are tested, and the results are used to optimize the operational parameters further to better suit each flight environment.

## 2. Methodology of Moving Object Detection from a Moving Camera

### 2.1. Projective Transformation

A transformation in image processing refers to a function that maps an arbitrary point to another point. A 2D transformation is a specific category of transformation, which performs 2D-to-2D mapping. The most general form of a 2D transformation is a projective transformation, which can be interpreted as a type of transformation that maps an arbitrary quadrangle onto another quadrangle. Thus, a projective transformation can describe the relationship between two images that show the same planar object. The matrix that performs a projective transformation is called homography matrix, and the transformation can be defined as follows:(1)$$\left[\begin{array}{c} x_{t}\\ y_{t}\\ 1 \end{array}\right]\sim \left[\begin{array}{ccc} h_{11} & h_{12} & h_{13}\\ h_{21} & h_{22} & h_{23}\\ h_{31} & h_{32} & h_{33} \end{array}\right]\cdot \left[\begin{array}{c} x\\ y\\ 1 \end{array}\right]$$
$$where\;\left[\begin{array}{c} x_{t}\\ y_{t}\\ 1 \end{array}\right]\;is\;the\;pixel\;coordinate\;after\;the\;transform.$$

A homography matrix has eight degrees of freedom, and at least four matches between images are required to compute this matrix. Generally, more than four matches are processed with outlier-rejection techniques such as RANSAC [21] or LMedS [22] to yield optimal results.

In principle, a projective transform can only describe planar, or 2D, objects. However, as shown in Figure 3, if 3D objects are relatively far away from the camera, and the posed differences between viewpoints are minor, the relationship between the images taken from these viewpoints can be approximated with a projective transformation. This is a core assumption of a moving object detection system. Details about how to utilize this assumption for background motion modeling is provided in the next section.

### 2.2. Feature-Point Matching between Images with an Optical Flow

The optical flow methodology estimates the motion vectors of pixels during two consecutive frames. It involves a calculation of a differential equation called the gradient constraint equation, and methodologies for solving this equation consist of various optical flow algorithms. Some of the popular methods are the Lucas–Kanade and Horn–Schunck algorithms. The Lucas–Kanade algorithm only tracks specified pixels but is computationally less complex. The Horn–Schunck algorithm tracks all pixels in a frame but with less accuracy and at lower speeds. For the proposed system, the Lucas–Kanade optical flow is employed to match pixels between consecutive image frames. A combination of epipolar geometry, projective transformation, and an optical flow enables the background motion modeling and moving object detection. The detailed procedures are presented in the following section.

### 2.3. Moving Object Detection from a Moving Camera

The mid-air collision avoidance system proposed here consists of four operational stages: detection, recognition, tracking, and decision-making. The detection stage detects regions of moving objects in the camera’s field of view. A pseudocode of this stage is given in Algorithm 1.

**Algorithm 1** Independent moving object detection**Input:** It−1,It,It+1,Ht−1,t**Output:** Rt**1:** FP ← 0 0 0 0 0 0**2:** i=0**3:** while i<6:**4:**     FP[i] ← number of feature points in Iti**5:**     if FPi<n:**6:**         Find at least n−FP[i] feature points in Iti**7:**     endif**8:** end while**9:** OFt,t+1← optical flow between Iti and It+1i           ► Track background feature points**10:** Ht,t+1 ← estimate homography between Iti and It+1i using OF►Calculate homography matrix**11:** Rt ← average([(It−Ht−1,t·It−1)+(It+1−Ht,t+1·It)])►Background subtraction**12:** Rt ← Binarize(Dilation(MedianFilter(Rt)))           ►Apply denoising filters and binarization

The purpose of homography matrix calculation between consecutive images is to model background scene transitions caused by camera motion. If three-dimensional objects are relatively far away, and the pose difference between the viewpoints is small, the transition between two video frames can be approximated as a projective transformation. Figure 4 visualizes this approximation process.

The calculated homography matrix is then used for background subtraction. This matrix approximately relates pixels from the previous frame to pixels of the current frame. That is, pixels from the previous frame can be mapped to coincide with those of the current frame. However, the motions of moving objects are not described by the homography matrix and thus cannot be mapped to coincide between frames. Therefore, they can be detected by searching for regions where pixels are not correctly mapped between consecutive frames. To locate incorrectly mapped pixels and detect moving objects, a perspective transform-based background subtraction technique is employed. The homography matrix between two consecutive frames is used to map pixels of the previous frame to match those of the current frame. The transformed frame is then subtracted from the current frame. The pixels of moving objects are not correctly mapped to the current frame given that the motion of the current frame differs from that of the background; accordingly, the corresponding brightness has a nonzero value after subtraction, signifying that image regions of moving objects are detected. An example result is shown in Figure 5. The size of the quadcopter drone used in the experiment was 180 × 65 × 190 mm. Thus, the proposed algorithm can detect a small-size moving drone.

Also, when calculating the homography matrix, the flight conditions of UAVs were taken into consideration. From the viewpoint of a flying UAV, objects near the center of the field of view are typically further away than those at the edge areas. Further objects appear smaller and thus exhibit weaker magnitudes of the appearance transition. Thus, objects near the edge of the field of view have more influence on the perspective transformation model. Therefore, it is reasonable to calculate the homography matrix from the transition of pixels that are close to the edge of the field of view. In this system, six 70 × 70 windows were used. The system constantly monitors the number of tracked points in these windows and initiates new tracks if the number falls below a predetermined threshold. This evenly distributes the number of tracked points and guarantees the stability of the homography matrix calculation. To filter out false matches and outliers, RANSAC was used to calculate the homography matrix. Limiting the optical flow calculation regions reduced the computational requirements and improved the approximation accuracy simultaneously. 

However, because the projective transformation approximation is not perfect, noise elements inevitably exist. That is, even after background subtraction, some areas of the background may not have zero brightness values. To compensate for this, the proposed system utilizes two methods. The first utilizes the next frame as well as the previous frame for background subtraction. Each transformed frame is subtracted from the current frame and then averaged. Because noise elements appear randomly and momentarily, this method improves the signal-to-noise ratio (SNR) of the resulting image. The second method applies image filters and binarization to the resulting image after the first method. A median filter eliminates small noise components, while a dilation filter expands elements, thus revealing moving object regions more clearly. The image is subsequently binarized to disregard pixels below a certain brightness threshold. This process is illustrated in Figure 6.

## 3. Methodology for Object Recognition, Tracking, and Decision-Making

### 3.1. Object Recognition

The recognition stage applies a clustering algorithm to the result of the tracking stage to determine the number and locations of all moving objects. A pseudocode of this stage is given in Algorithm 2.
**Algorithm 2** Independent object recognition**Input:** Rt**Output:** Ct**1:** Ct←∅**2:** for unallocatedBlob in Rt:► A region  in Rt**3:**
if sizeunallocatedBlob<sizeThresh:**4:**
     continue► Disregard small regions**5:**
for cluster in Clusters:**6:**
        for blob in cluster:**7:**
            if distanceblob, unallocatedBlob<distThresh:**8:**
                allocate unallocatedBlob to cluster**9:**
            break                   ► Add region to cluster if below distance threshold**10:**
    if unallocatedBlob is unallocated:**11:**
        allocate unallocatedBlob to Clusters and set as cluster► Set as new cluster**12:**
for cluster in Clusters:**13:** append weighed mean position of cluster to Ct       ► Append representative position of blob to Ct


The recognition phase outputs a binary image that displays the regions of moving objects as 1. However, as the presence of noise is inevitable, some background regions have a value of 1, albeit intermittently. Additionally, a single independent moving object can appear in several “patches”, as shown in Figure 7. Therefore, there is a need for a method capable of rejecting noise components while also recognizing multiple nearby patches as a single object.

For the proposed system, a modified DBSCAN (density-based spatial clustering of applications with noise [23]) algorithm is developed. The DBSCAN, unlike center-based algorithms such as K-means [24], determines if a data point belongs to a cluster based on its distances to other points. It regards data points with no neighbors as outliers and data points with more than some number of neighbors as inliers. However, a vanilla DBSCAN cannot be directly applied for this task, as there are instances in which an object shows up as a single patch as well as those where noise components appear as several patches. Thus, the vanilla DBSCAN is modified to utilize the area information of patches. Only patches within an area threshold are considered as a true object, and a single patch is considered as a cluster if it satisfies the area threshold. The centroid of a cluster is calculated using the weighed sum of all patches in that cluster. This procedure is summarized in Figure 7.

### 3.2. Object Tracking

The tracking stage receives the objects’ locations at each frame from the recognition stage and associates them with those of the previous frame. A unique ID is assigned to each independent object, and the corresponding trajectories are monitored for threat assessment. Additional noise compensation is also provided in this stage. A pseudocode of this stage is given in Algorithm 3.
**Algorithm 3** Independent object tracking**Input:** Ot−1,Ct**Output:** Ot**1:** O←∅**2:** M=size(Ot−1)                      ► number  of objects at previous frame**3:**  N=size(Ct)                       ► number of objects at current frame**4:** Array ←zeros(M, N)              ► An array that stores distances between all objects**5:** for i in Ot−1:**6:**
   for j in Ct:**7:**
        Arrayi, j=distance(Ot−1i, Ctj)**8:**
for k in Ct:**9:**
    if ID[k] isnot set:**10:**
     if Arrayi, k==min⁡(Arrayi, :):            ►  object *k*: closest object to object *i***11:**
         if  Arrayi, k<distThresh:                 ►  if below distance threshold:**12:**
             ID[k]←ID[i]              ► assume *k* and *i* is the  same object (tracking success)**13:**
            Tracked length of IDk+=1► Increase tracked length**14:**
            break**15:**
         else:**16:**
            mark ID[i] as lost                  ► Mark  as lost if no object is nearby**17:**
for k in Ct:**18:**
    if ID of k isnot set:**19:**
          assign new ID to k                ► assign  new *ID* if new object is detected**20:**
          tracked length of IDk←0**21:**
    if tracked length of IDk>noiseThresh:            ► If  tracked longer than threshold**22:**
          IDk←mark as true object► Approve as true object**23:**
    add position and ID of k to Ot


Temporal information is used to distinguish true observations from noise and to recognize object identities across multiple frames. This step operates on the central assumption that the distance travelled by an object between frames is shorter than the distances between separate objects. Moreover, because noise components are spawned intermittently and momentarily, they do not appear consistently across multiple frames.

First, the distances between previously detected coordinates and new coordinates are compared, after which the closest new detection within a distance threshold is recognized as the new position of the object. If there are no new detections within a distance threshold, the object is marked as lost. If no new detections appear near its last known location for a time threshold, the object is deleted. New detections with no associations are identified as object candidates and are given a temporary ID. If an object candidate is successfully tracked for a time exceeding a certain time threshold, it is approved as a true object and given a new ID. This procedure is summarized in Figure 8.

### 3.3. Decision-Making for Avoidance Maneuvers

The decision-making stage receives the trajectory data from the tracking stage and determines whether avoidance maneuvering must be performed, and if so, in what direction it should occur. A pseudocode of this stage is given in Algorithm 4, and Figure 9 presents a diagram of this algorithm.
**Algorithm 4** Decision-making for avoidance maneuvers**Input:** Ot                        ► *O_t_*: object IDs  and locations at current frame**Output:**
At                        ► At: object IDs  and locations at current frame**1:**
for object in Ot:**2:**
    position←3D position of object                  ►  acquire 3D position of object**3:**
if position.z<dstopThresh :**4:**
        emergency stop before proceeding**5:**
        break**6:**
    if position is inside RSafeWindow:**7:**
        mark object as hostile               ► consider object as  a threat if inside *SafeWindow***8:**
        if position.z<davoidThresh:      ► execute avoidance maneuver if  distance is below threshold**9:**
            if position.xy in upperRegion:**10:**
                move downward at v m/s until outside of RSafeWindow**11:**
            if position.xy in lowerRegion:**12:**
                move upward at v m/s until outside of RSafeWindow**13:**
            if position.xy in leftRegion:**14:**
                move right at v m/s until outside of RSafeWindow**15:**
            if position.xy in rightRegion:**16:**
                move left at v m/s until outside of RSafeWindow**17:**     ***break***


The proposed system uses the object trajectory information from the tracking system to perform avoidance maneuvers if the relative distance falls below a certain threshold. Additionally, if the initial measured distance to the object is too close (for example, if the moving object approached from the side of the vehicle outside the field of view), an emergency stop occurs to prevent a crash into the obstacle. The radius of the safe window and the avoidance threshold can be determined from prior information, such as the object’s expected maximum velocity and the maximum possible maneuver velocity.

## 4. Performance Validation of In-Flight Collision Avoidance

In this section, the performance validation of the proposed in-flight collision avoidance system will be demonstrated. Before the actual in-flight collision avoidance testing, in-flight dynamic object detection based on background subtraction was conducted. Then, a simulation test was conducted using ROS Melodic with Gazebo environment. After the simulation testing, in-flight dynamic object detection based on background subtraction was conducted. Lastly, the proposed system (in-flight collision avoidance) was tested with actual quadcopter drone. 

### 4.1. System Setting for In-Flight Collision Avoidance

The UAV hardware setting for the proposed in-flight collision avoidance shown in Figure 2. It is equipped with a Raspberry Pi 4 companion computer, a RealSense D435i stereo depth camera, and a RealSense T265 visual odometry camera for an indoor autonomous flight. The system components are interconnected via ROS (Robot Operating System), a robotics development software package, as illustrated Figure 10. 

### 4.2. Validation of Moving Object Detection and Tracking

Before the actual in-flight collision avoidance testing, the in-flight dynamic object detection based on background subtraction was conducted as demonstrated in Figure 11. As Figure 11 shows, the in-flight UAV was able to subtract the dynamic background and detect only the moving object. 

Furthermore, the detection and tracking algorithm is based on the Kalman filter. The tracking results for video sequences are shown in Figure 12. The IDs of objects are assigned to independent objects, and the two-dimensional trajectory of the object is displayed as red points. The video result is available at https://youtu.be/AcWcUMl0WW8 (accessed on 20 June 2023). The tracking algorithm works in real time on a Raspberry Pi 4 computer. It compensates for noise components and keeps track of the objects even in the detection failure.

The performance validation for moving object detection and tracking algorithms was also conducted for real-time performance. The fps (frames-per-second) performance comparison results with FastMCD [15] and a neural-network-based system [25] are displayed in Figure 13. The blue line represents FastMCD, the orange line represents the neural-network-based system, and the gray line represents the proposed system. As shown in the figure, the proposed system is vastly and continuously superior in terms of real-time processing performance, processing up to 70 fps and 60 fps on average. Note that the stereo camera system utilized for the collision avoidance system streams video data at 30 frames per second, and the proposed algorithms always performed sufficiently above the real-time requirements for the task of in-flight moving object detection, tracking, and collision avoidance. The validation demonstrated the real-time suitability for in-flight collision avoidance tasks in real time.

### 4.3. Validation of In-Flight Collision Avoidance

Before the actual in-flight testing, a simulation test was conducted using ROS Melodic with Gazebo environment as shown in Figure 14. 

After the simple simulation test, the operational variables were set for the actual in-flight collision avoidance. To validate the proposed system, the drone was set to fly forward until avoidance maneuvers were required. The operational variables were set as listed in Table 1.

For the first and second validation tests, the emergency stops were demonstrated. The drone detected an obstacle closer than 0.5 m away and performed the emergency stop before proceeding. These results are shown in Figure 15, Figure 16 and Figure 17 with plots of the trajectories of the drone and the object. Table 2 is the list of various detection results and the avoidance performance metrics during in-flight performance. For the performance validation of the proposed in-flight avoidance system, a ball with a size of 200 mm was thrown to the drone.

For the third and fourth validation tests, the avoidance maneuvers were demonstrated. The vehicle detected and tracked an obstacle from outside the safe window and performed avoidance maneuvers when the object breached the avoidance threshold. These results are shown in Figure 18, Figure 19, Figure 20 and Figure 21. Figure 19 presents a plot of the trajectories of the vehicle and the object and the nearest distance between them at the same point in time. Table 3 is the list of various detection results and avoidance performance metrics during the process.

Figure 22 presents the detection, recognition, and tracking results from the UAV’s point of view. The red points represent the tracked two-dimensional trajectory of the object. 

To demonstrate the effectiveness of our approach, we compare some of our test results to those of [25,26], which describes an event-camera-based obstacle avoidance system. Although the types of sensors used for each systems differ, the components of the algorithms are very similar: ego-motion compensation, thresholding, morphological operations, and clustering. The algorithm in [25] runs much faster than ours—up to 280 fps—thanks to a simpler ego-motion compensation process that is made possible with the characteristics of the event camera, whereas our approach is bottlenecked by feature point extraction and homography computation processes. Nevertheless, the results for indoor obstacle avoidance for each method are comparable; the approach in [25] was analyzed to enable the avoidance of dynamic obstacles with relative speeds of up to 10 m/s, with reactions times of below 0.2 s. Our approach was tested and demonstrated with dynamic obstacles at relative speeds of up to 6 m/s with similar reaction times. We presume that with a higher-powered computer, such as the NVIDIA Jetson TX2 used in [25], our proposed system would be able to process higher-resolution images at a constant time rate, which would certainly help to detect the objects that are further away or move with higher relative speeds.

However, on several aspects of the current system can improve the low-cost in-flight collision avoidance. First, the proposed background subtraction algorithm must be improved. The proposed algorithm cannot deal with the drastic light changes. The object detection should be robust in various light-interference conditions (i.e., dawn, flying conditions with the sun high in the sky, etc.). To overcome those critical challenges and achieve robust real-time object detection, light-weight CNN-based algorithm should be implemented. Second, the in-flight collision avoidance algorithm must be improved. This research is focused on development of low-cost vision-based moving obstacle detection on the moving UAV. Thus, the probabilistic decision-making algorithm must be implemented. Tutsoy [27] proposed that parametric machine learning algorithms have been developed not only to predict the future responses of the systems but also to produce constrained policies (decisions) for optimal future behaviors under the unknown uncertainties. 

Future work will attempt not only to extend the proposed in-flight collision avoidance system to outfield validation using multiple moving objects in various light conditions but also to extend the computing system for deep-learning-based or probabilistic real-time decision-making. 

## 5. Conclusions

This paper proposed a high-performance and low-cost in-flight collision avoidance system based on background subtraction for unmanned aerial vehicles (UAVs). A novel vision-based mid-air collision avoidance system for UAVs was proposed and implemented. The pipeline of the proposed in-flight collision avoidance system is as follows: (i) subtract dynamic background to remove it and to detect moving objects, (ii) denoise using morphology and binarization methods, (iii) cluster the moving objects and remove noise blobs, using Euclidean clustering, (iv) distinguish independent objects and track the movement, using the Kalman filter, and (v) avoid collision, using the proposed decision-making techniques. Performance validation tests were conducted in a simulation environment and with an actual quadcopter drone with vision sensors. 

The key contribution of the proposed system is a lightweight, error-compensating moving object detection and recognition algorithm. Specifically, background scene transitions due to ego-motion are approximated with a two-dimensional projective transformation, and a homography matrix is computed using the optical flow. To reduce the required computational load and improve the quality of approximation, the optical flow is calculated only at the edge regions of video frames. The previous and successive frames are transformed to match the current frame, and the background subtraction is performed to acquire a primitive estimate of the object location. Image filters and thresholding are utilized to improve the SNR of this result. A modified DBSCAN clustering algorithm is used to identify multiple detected image patches correctly as a single object. A distance-based tracking algorithm assigns object identities and tracks them across frames, incorporating an additional noise-filtering procedure based on the tracked period. A stereo camera system measures the distances to detected objects, and this information is used for determining whether avoidance maneuvers must be executed. 

Further research on several aspects of the current system can improve low-cost in-flight collision avoidance. Future work will attempt to develop an object classification system for greater reliability and robustness of the system by developing a light CNN with a six-degree-of-freedom pose estimation algorithm to detect an orientation of moving objects. 

The proposed system effectively detects, recognizes, and tracks moving objects in its field of view with low computational requirements and achieves sufficient processing performance on a low-power onboard computer. It is implemented onto a test vehicle and performs mid-air collision avoidance to demonstrate its effectiveness in real-world conditions.

## Figures and Tables

**Figure 1 sensors-23-06297-f001:**
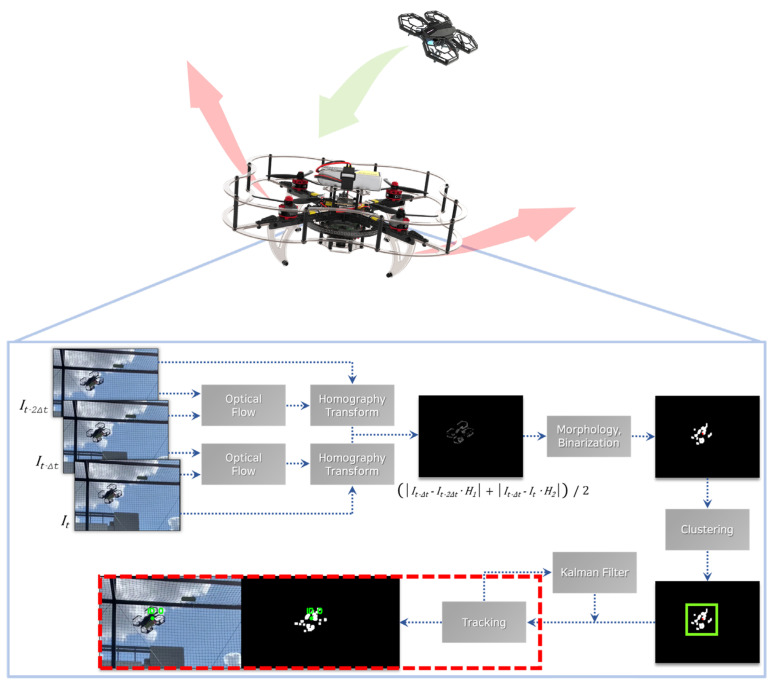
Concept of applying the proposed low-cost moving object avoidance system.

**Figure 2 sensors-23-06297-f002:**
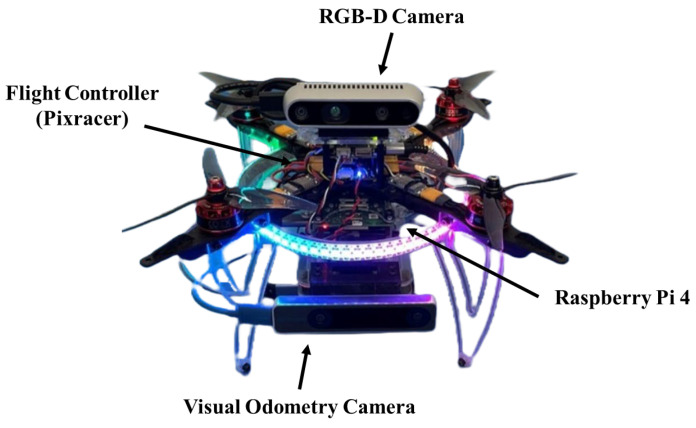
Proposed hardware setting for the low-cost moving object avoidance.

**Figure 3 sensors-23-06297-f003:**
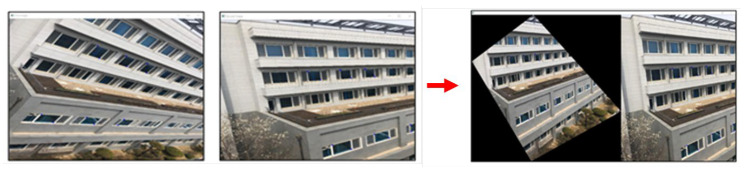
Approximation and transformation of background change between two viewpoints with projective transformation.

**Figure 4 sensors-23-06297-f004:**
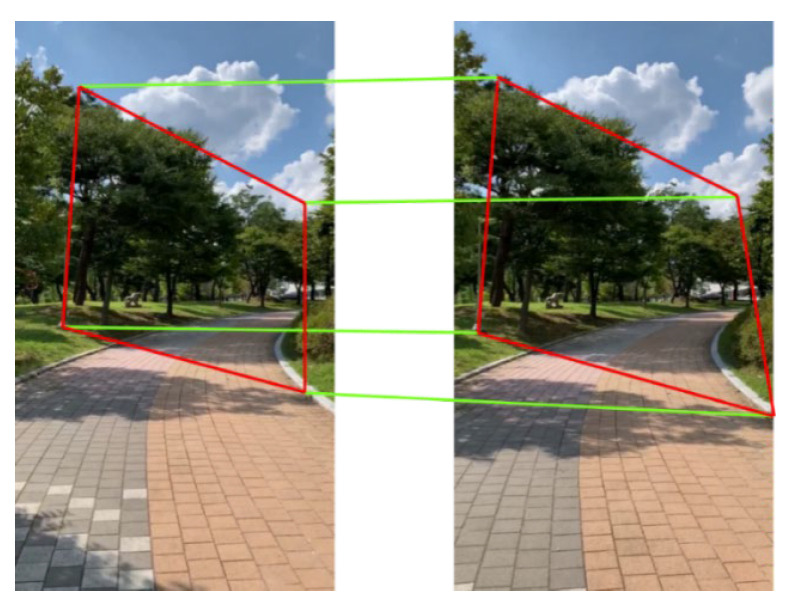
Projective transformation approximation of two consecutive video frames.

**Figure 5 sensors-23-06297-f005:**
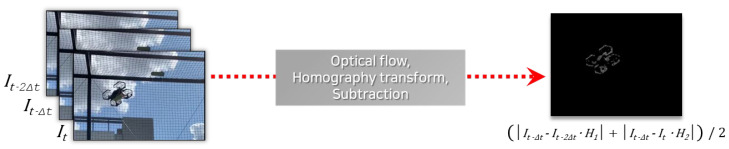
Original image and background-subtracted result.

**Figure 6 sensors-23-06297-f006:**
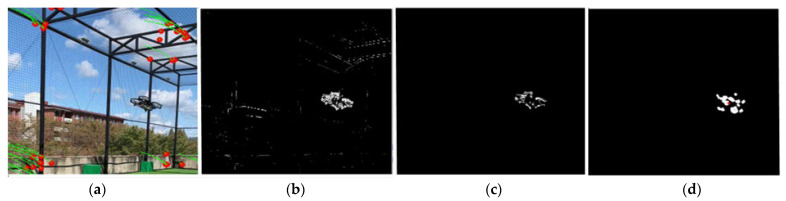
Image processing for background subtraction: (**a**) the original frame, (**b**) background subtracted frame, (**c**) median-filtered frame, and (**d**) dilated and binarized frame.

**Figure 7 sensors-23-06297-f007:**
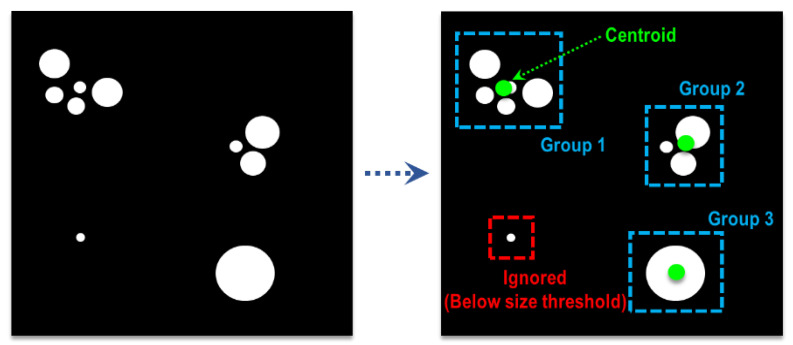
Clustering procedure of the modified DBSCAN algorithm.

**Figure 8 sensors-23-06297-f008:**

Tracking procedure of the distance-based tracking algorithm.

**Figure 9 sensors-23-06297-f009:**
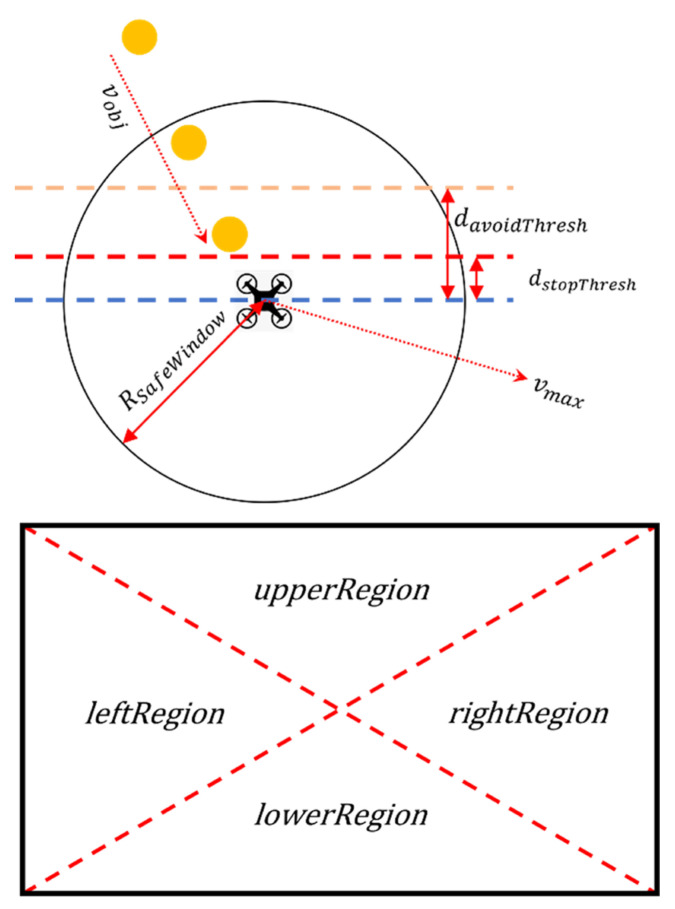
Illustration of avoidance maneuvers.

**Figure 10 sensors-23-06297-f010:**
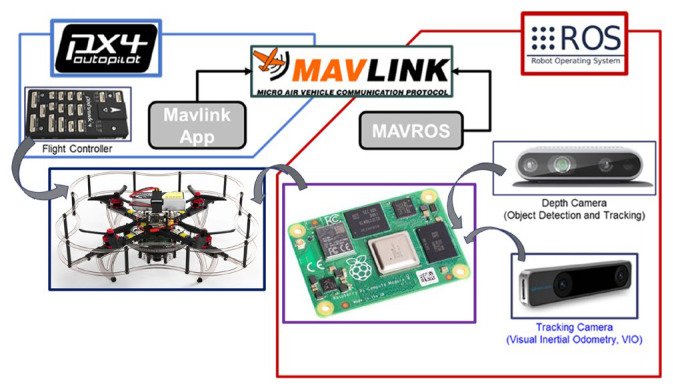
Software and hardware integration for the proposed system.

**Figure 11 sensors-23-06297-f011:**
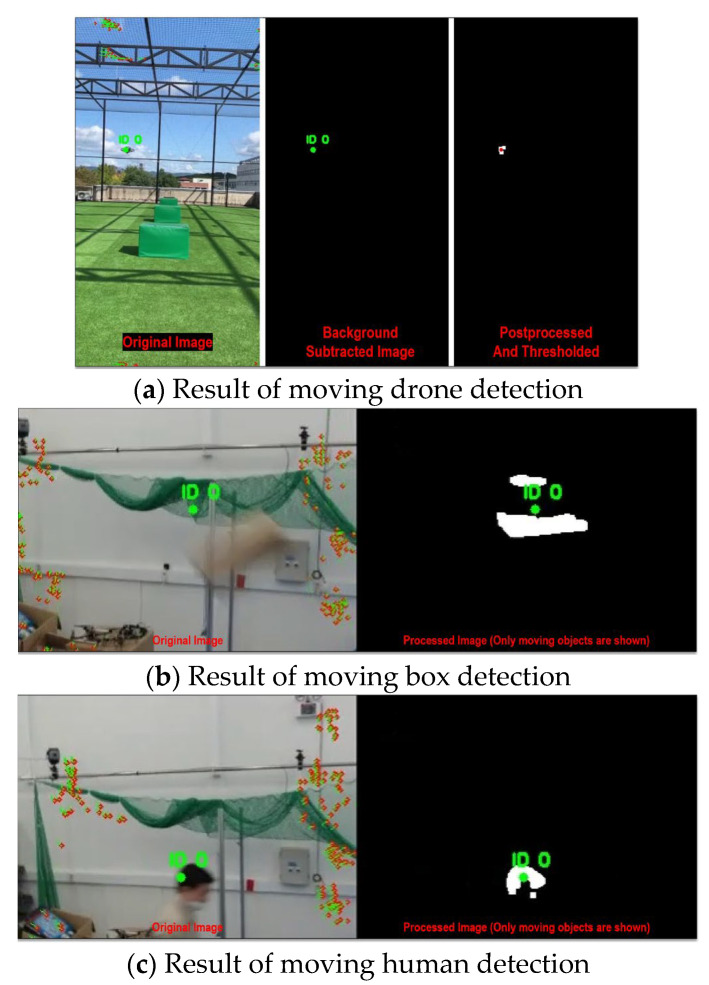
Moving object detection based on background subtraction.

**Figure 12 sensors-23-06297-f012:**
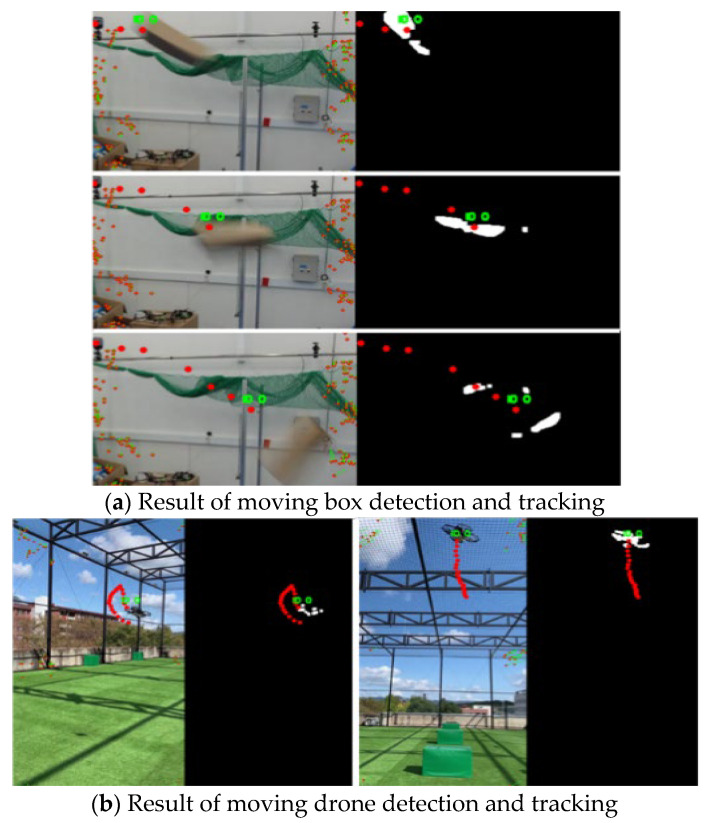
The results of moving object detection and tracking.

**Figure 13 sensors-23-06297-f013:**
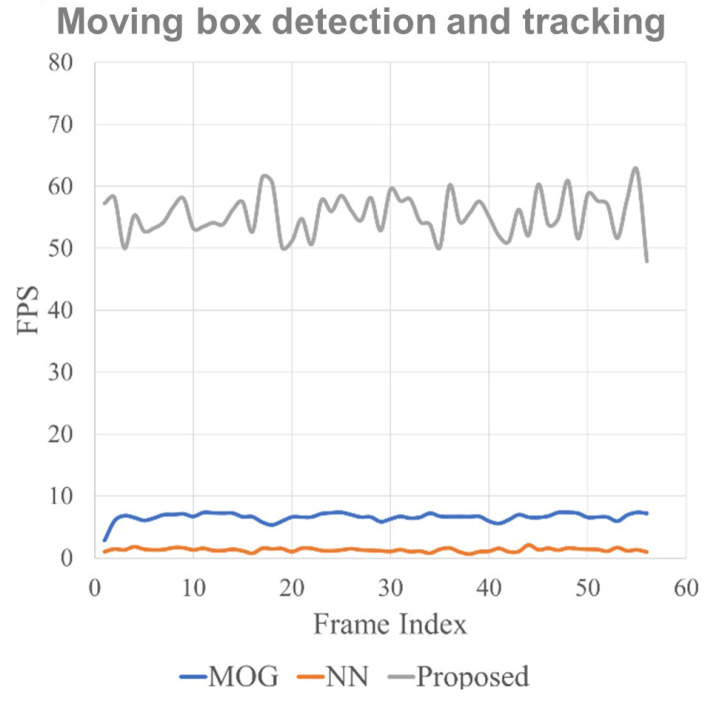
The comparison of fps processing performance with other moving object detection systems.

**Figure 14 sensors-23-06297-f014:**
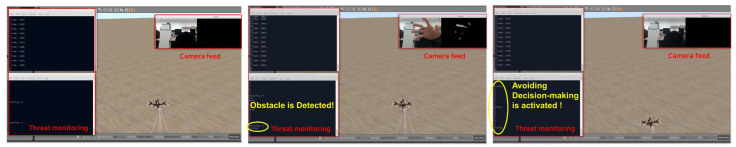
Demonstration of the in-flight collision avoidance in ROS Gazebo simulation environment.

**Figure 15 sensors-23-06297-f015:**
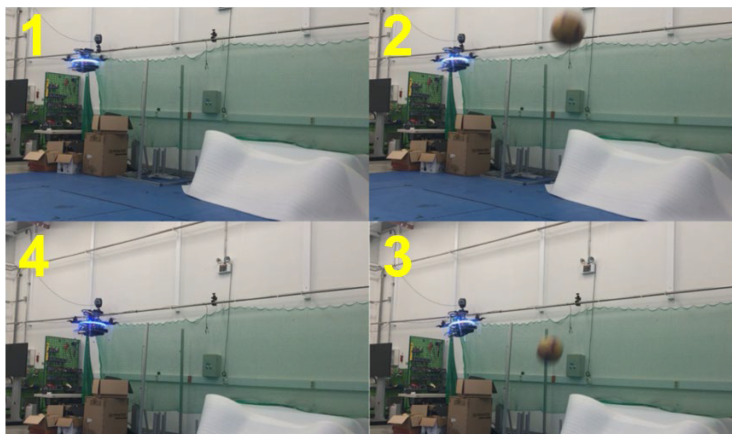
Demonstration of emergency stop 1.

**Figure 16 sensors-23-06297-f016:**
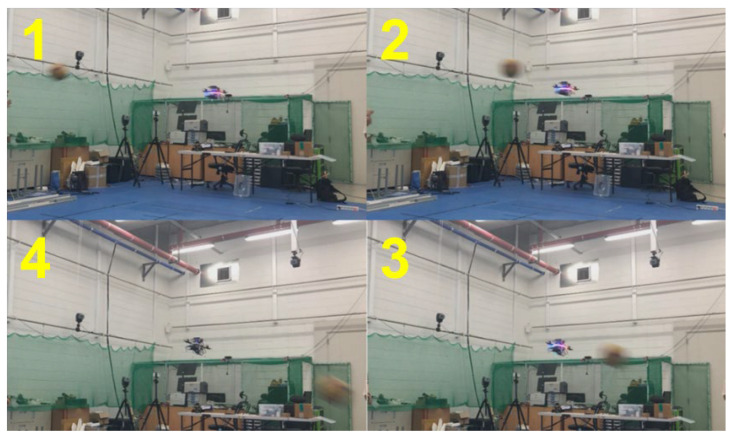
Demonstration of emergency stop 2.

**Figure 17 sensors-23-06297-f017:**
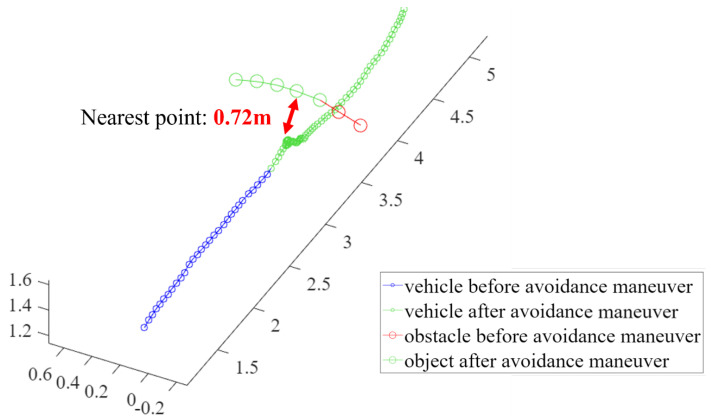
The trajectories of the drone and the object.

**Figure 18 sensors-23-06297-f018:**
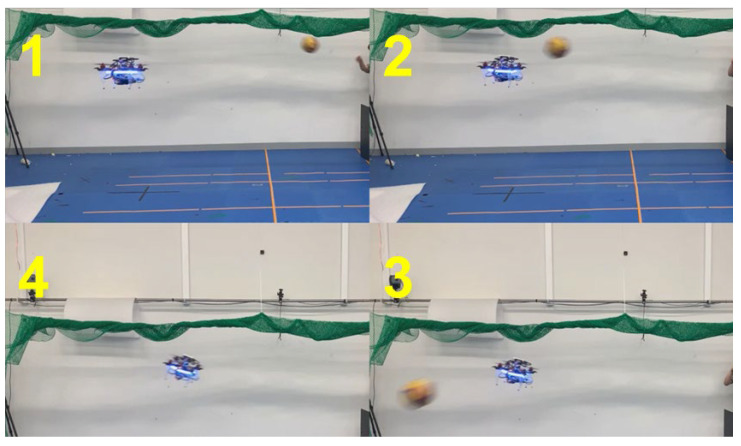
Demonstration of avoidance maneuver 1.

**Figure 19 sensors-23-06297-f019:**
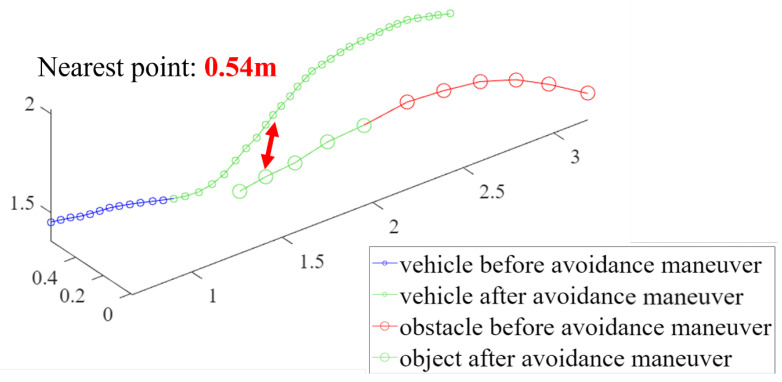
The trajectories of the drone and the object for avoidance maneuver 1.

**Figure 20 sensors-23-06297-f020:**
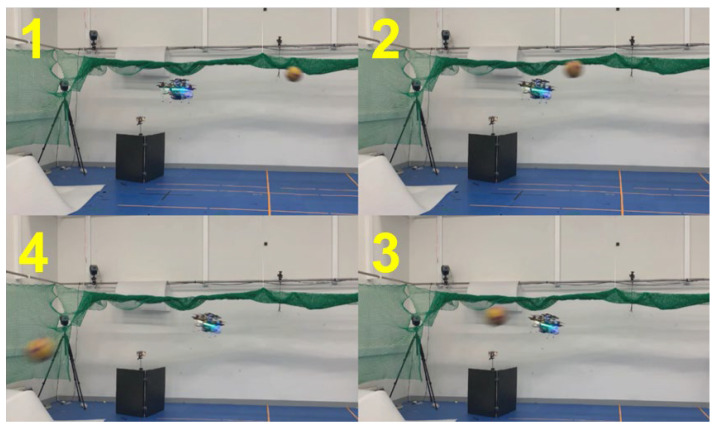
Demonstration of avoidance maneuver 2.

**Figure 21 sensors-23-06297-f021:**
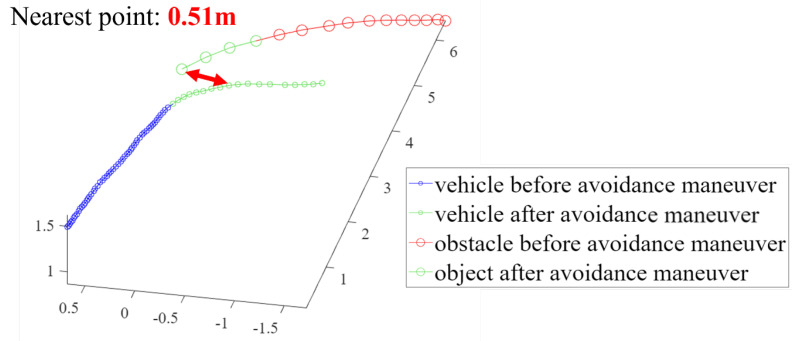
The trajectories of the drone and the object for avoidance maneuver 2.

**Figure 22 sensors-23-06297-f022:**
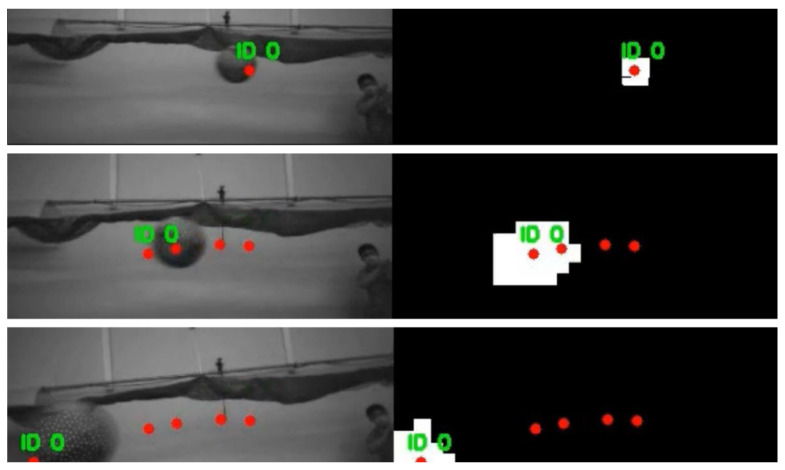
Detection, recognition, and tracking results from UAV′s point of view.

**Table 1 sensors-23-06297-t001:** Operational variables for the in-flight collision avoidance.

Algorithm 1 Independent moving object detection	
Parameter	Value
image resolution	424 × 240
boundary region size	70 × 70
Number of regions	10
median filter size	5 × 5
dilation filter size	5 × 5
binarization threshold	40
Algorithm 2 Independent object recognition	
Parameter	Pixel
sizeThresh	25 px
distThresh	100 px
Algorithm 3 Independent object tracking	
Parameter	Value
distThresh	150 px
noiseThresh	
Algorithm 4 Decision-making for avoidance maneuvers	
Parameter	Value
$R_{SafeWindow}$	1 m × 1 m × 1 m
$d_{avoidThresh}$	1 m
$v_{max}$	3 m/s
$d_{emergencyThresh}$	1 m

**Table 2 sensors-23-06297-t002:** The detection results and avoidance performance metrics.

Emergency Stop	
Category	Value
Vehicle speed	1.2 m/s
Obstacle speed	5.9 m/s
Relative speed	6.0 m/s
Time between detection and recognition	0.09 s
Minimum distance to obstacle	0.72 m

**Table 3 sensors-23-06297-t003:** The detection results and avoidance performance metrics.

Avoidance Maneuver 1	
Category	Value
Vehicle speed	1.2 m/s
Obstacle speed	5.7 m/s
Relative speed	6.7 m/s
Time between detection and recognition	0.1 s
Minimum distance to obstacle	0.54 m
Avoidance Maneuver 2	
Category	Value
Vehicle speed	1.2 m/s
Obstacle speed	5.6 m/s
Relative speed	5.9 m/s
Time between detection and recognition	0.11 s
Minimum distance to obstacle	0.51 m

## Data Availability

Not Applicable.

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
