# Peer review of "Vision-Based In-Flight Collision Avoidance Control Based on Background Subtraction Using Embedded System"

_sensors, 2023, doi:10.3390/s23146297_

Round 1

Reviewer 1 Report

Overall, the article presents a necessary concept of low cost UAV collision avoidance system and is a great start, using simple image processing techniques couple with vision based ML. The work is presented in a way that is easy to follow with pictorial and video explanations along the way. The language used is clear to understand as well.

However, there a couple of points to think about, could be elaborated in the article if the authors wish so -

1. It would be great if there were examples of multiple moving objects simultaneously and how the detection algorithm adapts to these situations. If it is a current limitation, it is fine to put it as such.

2.  Does the brightness conditions of the background have an impact on object detection, if so, it would be great to elaborate as a limitation or opportunity to improve.

3. Would there be a size range for objects to be detected (small objects such as birds etc.) or would it be expected to work with similar level of accuracy as with a large object such as basketball or another drone?

4. There should always be a section on current limitations in an article, it seems like this is an area that can be improved before publication.

Reviewer 2 Report

- Figure 1 should be reorganized focusing especially on the diagram given on the right side of the figure. They are not readable and understandable. A similar problem exists in words located left side of Figure 9.

- In the paragraph starting from Line 82, the literature survey given about the methodologies for moving object detection should be extended with more studies, and each methodology’s success, difficulties, or advantages should be mentioned.

- In line 117, there is the word “with” used different font type incorrectly.

- Table 1, 2, and 3 looks confusing and should be reorganized with vertical and horizontal separator lines.

Should be improved slightly.

Reviewer 3 Report

The paper concerns with design of a vision-based fast moving object detection and tracking. Its focus is important, but the main body of the paper has not been properly constructed to deliver required contributions of the paper. The results should be enhanced as well. Please consider the comments below to improve the paper further:

1)      It is stated that the perception has high performance and low-cost. Please show and prove these properties clearly.

2)      Detecting the moving objects depends on a number of properties. Please clearly state these properties and state how they are handled in this research.

3)      Decision making for collision avoidance is not clearly expressed and justified in the paper. What are the probabilities of the made decisions? How are the uncertainties incorporated into the detection problem?

4)      Please improve Algorithm 1 and 2. Please check the abbreviations, either they are not properly introduced or introduced for a couple of times. Please also improve the figures.

5)      Please provide more accurate and informative title for the paper. The title should reflect the key contributions of the paper. In the current form, it is too long and not quite strong.

6)      Abstract of the paper should be improved. The first sentence can state the importance of the content, then the gaps in the corresponding literature. Key contributions of the paper should be expressed clearly and then the major findings of the paper should be provided.

7)      Introduction has provided some background researches and highlighted their advantages and disadvantages. However, critical review of the recent and related works are not quite strong. The corresponding gaps should be emphasized strongly and based on these gaps, the claimed contributions of the paper should be justified.

8)      Performing a comparison-based analyses with a recent and related approach under the equal conditions could help to improve and justify the contribution of the paper.

9)      Please improve the equations by adding brief insights about them. In the current form, they are quite raw and not clear how they serve the purpose of the paper.

10)  Please specify the kind of uncertainties. They can be internal or external, parametric or non-parametric, constant, characteristic or random. Determining their structures and amounts are challenging in the real time applications.

11)  Paragraph main sections and sub-sections of the paper should be reorganized to enhance the presentation of the work.

12)  Please note that recently advanced parametric machine learning algorithms have been developed for both predict the future responses of the systems and also produce constrained policies (decisions) to produce optimal future behaviours under the unknown uncertainties. Such improvements can be addressed as well. You can see this recent and related one. Covid 19 epidemic and opening of the schools: Artificial intelligence based long-term adaptive policy making.

13)  What are the possible problems that the proposed algorithm can face in real time applications? What are the constraints which are unavoidable in real time environments? 

Good luck with the improvements…

The paper concerns with design of a vision-based fast moving object detection and tracking. Its focus is important, but the main body of the paper has not been properly constructed to deliver required contributions of the paper. The results should be enhanced as well. 

Reviewer 4 Report

The manuscript titled “A Low-Cost Perception and Control System for In-Flight Collision Avoidance of UAVs based on Background Subtraction” presents an algorithm that uses a set of image-processing tasks to develop mid-air collision avoidance  system using cameras. The authors present a system that also has capabilities of perception of moving objects with decision-making capabilities for performing avoidance maneuvers.

The research presents advancement in the field of autonomous flight systems with a methodology for mid-air collision avoidance that uses background subtraction. However, one of the major limitations is that the lighting conditions may affect the efficacy of the proposed methodology. However, compared to other systems, the proposed methodology has cost advantages and can be tested further for mass deployment.

The image processing tasks proposed by the authors to accomplish the tasks is robust and the manuscript presents an advancement in the field of mid-air collision avoidance system using cameras.

Round 2

Reviewer 3 Report

The paper has been revised as requested and can be accepted. 

Minor polishing is required.